# Multi-Classifier Fusion Based on MI–SFFS for Cross-Subject Emotion Recognition

**DOI:** 10.3390/e24050705

**Published:** 2022-05-16

**Authors:** Haihui Yang, Shiguo Huang, Shengwei Guo, Guobing Sun

**Affiliations:** 1College of Electronic Engineering, Heilongjiang University, Harbin 150080, China; 2201678@s.hlju.edu.cn (H.Y.); 2201766@s.hlju.edu.cn (S.H.); 2211849@s.hlju.edu.cn (S.G.); 2Key Laboratory of Information Fusion Estimation and Detection, Harbin 150080, China

**Keywords:** emotion recognition, Multi-Classifier Fusion, mutual information, SFFS, cross-subject

## Abstract

With the widespread use of emotion recognition, cross-subject emotion recognition based on EEG signals has become a hot topic in affective computing. Electroencephalography (EEG) can be used to detect the brain’s electrical activity associated with different emotions. The aim of this research is to improve the accuracy by enhancing the generalization of features. A Multi-Classifier Fusion method based on mutual information with sequential forward floating selection (MI_SFFS) is proposed. The dataset used in this paper is DEAP, which is a multi-modal open dataset containing 32 EEG channels and multiple other physiological signals. First, high-dimensional features are extracted from 15 EEG channels of DEAP after using a 10 s time window for data slicing. Second, MI and SFFS are integrated as a novel feature-selection method. Then, support vector machine (SVM), k-nearest neighbor (KNN) and random forest (RF) are employed to classify positive and negative emotions to obtain the output probabilities of classifiers as weighted features for further classification. To evaluate the model performance, leave-one-out cross-validation is adopted. Finally, cross-subject classification accuracies of 0.7089, 0.7106 and 0.7361 are achieved by the SVM, KNN and RF classifiers, respectively. The results demonstrate the feasibility of the model by splicing different classifiers’ output probabilities as a portion of the weighted features.

## 1. Introduction

Emotions are a person’s comprehensive expression of external stimuli. In our daily life, emotions play a vital role in communication, decision making, perception and artificial intelligence. Emotion recognition has received extensive attention, especially in human–computer interaction (HCI) [1,2]. This is because people’s needs for interpersonal communication are increasing. The main purpose of emotion recognition is to detect and simulate human emotions during HCI [3]. Emotions can be recognized in many ways, including facial expressions, speech and some physiological signals [4,5]. Facial expressions can be used for emotion recognition mainly because the muscle movements and expression patterns of the human face change due to different emotions [6]. For example, when people are happy, their faces are mostly smiling, and the corners of their mouths tend to rise [7]. When people are angry, they open their eyes subconsciously. Some researchers have also used thermal images for facial expression recognition [8,9]. This is mainly because the infrared heat map can reflect the distribution of the surface temperature of the object. Emotion recognition using speech is mainly based on the language expression in different emotional states. For example, people generally speak faster when they are in a good mood. Facial expression and speech recognition have the advantages of simple operation and a low cost. However, facial expressions and speech can very easily be deliberately concealed and do not always accurately express real emotions [10,11]. Compared with facial expressions and speech signals, physiological signals can obtain real data and cannot be faked. Physiological signals mainly include electrocardiogram (ECG), electromyogram (EMG), electroencephalogram (EEG) signals. However, compared with EEG signals, other physiological signals produce fewer differences and change at a slower rate under different emotions. As a result, they are greatly limited in real-time emotion recognition. An EEG is a medical imaging technique that reads the electrical activity produced by brain structures. It can better express the differences between various emotions and has a higher time resolution. In addition, multi-modal information can also be applied to identify emotions, and this method can achieve high accuracy. However, it is greatly restricted in practical applications because the effect in an open environment is not stable. Therefore, most researchers use EEG signals for emotion recognition.

However, in the process of research, most researchers focus on subject-specific situations. This kind of system has poor generalizability, because different people respond differently to the same stimulus. This problem was raised for the first time by Picard et al. [12]. Therefore, we need to build a specific model for each person to determine the user’s emotional state in HCI. Obviously, this is very time-consuming and requires a great computational cost. Then, we hope to find the commonalities between the emotions of different people, so as to realize a general model to predict emotions. This is also a problem that researchers urgently need to solve—cross-subject emotion recognition [13,14]. It has always been very difficult to identify cross-subject emotions based on EEG signals, because the generalizability of features among different subjects is very poor.

In previous studies, many scientists have compared the results of cross-subject experiments with the results of subject-specific experiments. For example, Kim et al. used a bimodal data fusion method and apply linear discriminant analysis (LDA) for emotion classification. LDA is a supervised algorithm that can be used not only for dimensionality reduction, but also for classification. In this way, an accuracy of 92% in subject-specific experiments could be achieved, but the best recognition accuracy among the three subjects was only 55% [15]. Zhu et al. extracted and smoothed the differential entropy features. The average classification accuracy rate across subjects was 64.82%, but the subject-specific accuracy rate could reach 90.97% [16]. Later researchers have used many methods to improve the detection accuracy of cross-subject emotions. For example, Zhuang et al. proposed a method of feature extraction based on empirical mode decomposition (EMD), which decomposes EEG signals into several intrinsic modal functions. The final classification accuracy rate was 70.41% [17]. However, some researchers have discovered that EMD is prone to mode mixing. Thus, ensemble empirical mode decomposition (EEMD) and a succession of other methods were subsequently proposed. Candra et al. used the wavelet entropy of the signal segment with a period of 3–12 s and achieved an average accuracy of 65% for valence and arousal [18]. Later, Mert and Akan et al. proved that using Hjorth parameters and correlation coefficients as features achieved better classification performance [19]. Li and Lu et al. tested the recognition performance of event-related desynchronization (ERD)/event-related synchronization (ERS) features extracted from different frequency bands and found that the best frequency band for identifying positive and negative emotions was 43.5–94.5 Hz, which is the higher gamma frequency band [20]. Lin et al. extracted differential asymmetry features to prove that frontal lobe electrodes and parietal lobe electrodes contain rich emotional information [21]. Rational asymmetry (RASM) and differential asymmetry (DASM) features are used to represent differences between the left and right hemispheres of the brain. RASM can be defined as the ratio of left and right brain power, and DASM is the difference between left and right brain power. Zheng et al. focused on finding stable cross-subject characteristics and fragments of neural EEG patterns for emotion recognition [22]. They found that the lateral temporal areas activated more for positive emotions than negative emotions in the beta and gamma bands, and the cross-subjects EEG characteristics mainly came from these brain areas and frequency bands. The accuracy of emotion classifiers based on EEG signals is still limited due to the individual differences in EEG signal characteristics. Researchers have applied information theory to emotion recognition. Using this strategy, information can be measured via entropy and mutual information. Entropy is used to measure the amount of information contained in things, while mutual information is used to measure the correlation between two sets of events [23,24]. It is worth noting that information theory has begun to be applied to computational biology. All kinds of information are continuously produced in the living body, and normal physiological activities are realized under the control and regulation of this information. Entropy can be used to measure the complexity of time series. With the further development of concepts and theories, entropy features have been widely used as new nonlinear dynamic parameters, such as sample entropy (SampEn), approximate entropy (ApEn), differential entropy (DE) and wavelet entropy (WE) [25]. Furthermore, since the mutual information can be used to evaluate the correlation between features and labels in emotion recognition, researchers have tried to use mutual information for feature selection [26].

Previous researchers mostly used machine learning methods. Later on, Zheng and Lu et al. used the differential entropy feature of multi-channel EEG data to train deep belief nets (DBNs), achieving an average accuracy of 86.08% [27]. Compared with machine learning, deep learning can indeed achieve better classification results. However, it is also accompanied by a series of problems, such as the need for a large amount of training data and higher requirements for hardware devices. Later, Yang showed that the poor generalizability of EEG signals between different subjects is due to the existence of redundant features [28], and he aimed to find an effective feature set to improve generalizability. He extracted a total of 10 linear and nonlinear features to form high-dimensional features and adopted the cross-subject emotion recognition method of combining a significance test (ST) and sequential backward selection (SBS) to select features. When the SBS algorithm selects features, it starts from the feature set and removes one feature from the feature set at a time, so that the evaluation function value is optimal after removing this feature. Finally, the results of the cross-subject emotion recognition in DEAP reached 72%. However, Yang used all 32 channels and all frequency bands of physiological signals in his experiments, which also resulted in a huge amount of calculation.

This research offers the following main improved features:(1)Through studying the mechanisms of the human brain and drawing on the experience of previous studies, only 15 EEG channels of the DEAP dataset and three frequency bands (alpha, beta and gamma) were selected for research. This could reduce the amount of calculation to a certain extent.(2)The accuracy of cross-subject emotion recognition has always been affected by the poor generalizability of features. We used a 10 s time window for data slicing to augment the sample, which could better train the model. In order to better characterize the EEG characteristics of each person, the high-dimensional features of the EEG signal in the time domain, frequency domain and nonlinear domain were extracted.(3)By drawing on the idea of the attention mechanism, we propose a Multi-Classifier Fusion method based on traditional machine learning classification. Before using multi-classifier fusion, the method of mutual information (MI) and sequential forward floating selection (SFFS) was adopted to select optimal features, which can improve the efficiency of the algorithm and reduce the impact of redundant features on the classification accuracy. Multi-Classifier Fusion can make full use of the probability outputs from classifiers and uses them as a weight feature for recursive prediction. In order to verify the effectiveness of Multi-Classifier Fusion, the hold-out and leave-one-out method were adopted to verify its performance based on the DEAP dataset.

## 2. Materials and Methods

The traditional EEG analysis process mainly includes EEG signal preprocessing, feature extraction, feature selection and emotion classification. In this paper, we combine two different feature selection methods to modify the features, as shown in Figure 1; this is the overall method proposed in our research. Furthermore, the proposed method (Multi-Classifier Fusion) is used to further improve the classification performance. Both the hold-out and leave-one-out method are used for validation in this work.

### 2.1. DEAP

In order to ensure that this work can be repeated and compared with other studies, we used the public DEAP dataset and obtained EEG signals from it [29]. DEAP was established by scholars such as Sander at Queen Mary University of London. This dataset is a multi-modal open dataset. It is used for analyzing emotional states based on human physiological signals. This is also a unique dataset that uses music videos as emotional stimuli. The data collector uses the MP150 multi-channel physiological signal recorder to record. There are a total of 40 channels of physiological signals, including 32 channels of EEG signals and 8 channels of peripheral signals. After watching the music video for up to 1 min, the participants evaluate self-assessment manikins (SAM) for five aspects: valence, arousal, control, likeness and familiarity. Participants rate these four dimensions on a continuous scale between 1 and 10. The preprocessed EEG samples were used for this research, with a sampling frequency of 128 Hz and a frequency range of 4–45 Hz.

### 2.2. Data Preprocessing

#### 2.2.1. Data Preprocessing

In the experimental data collection, each subject watched a 60 s video. Considering that it takes a certain amount of time for a subject to enter an emotional state, the first part of the collected physiological signals could not accurately represent specific emotions, so only the last 40 s of EEG data were studied. In addition, considering that enough samples are needed in cross-subject emotion recognition, we tended to cut data samples by selecting a time window to amplify samples. In EEG-based emotion recognition, the width of the time window affects the classification results. Candra et al. used the DEAP dataset to investigate the effect of window size on emotion classification [18]. They experimented with time windows of different widths. Studies have found that better results can be achieved with window lengths of 3–12 s [30]. Based on their conclusion, the width of the time window was set to 10 s in the current work. Following this time window determination, the 40 s of EEG data were divided into four 10 s EEG sections. Hence, for each person, we could acquire 160 samples (40 videos*4 sections); with a total of 32 participants, a total of 32 × 160 = 5120 samples were obtained.

#### 2.2.2. Label Processing

For the DEAP dataset, we could obtain 5120 (32*160) experimental samples. In this study, we determined the label according to the score of the participant. It was initially decided that if the participant’s score was less than or equal to 5, the label was set to 0. If the score was greater than 5, the label was set to 1. In short, we considered a binary classification for the valence dimension.

#### 2.2.3. Channel Selection

DEAP contains 32 channels of signals, which are distributed across four areas of the brain. These include the frontal lobe, parietal lobe, occipital lobe and temporal Lobe. Different areas have different functions [31]. However, researchers have shown that not all channels are indispensable in the process of emotion recognition. If all channels are used, it will inevitably lead to the problem of feature redundancy and a large amount of calculation [32].

In the study of emotion recognition, researchers have explored the optimal combination of EEG channels for feature recognition and have tried a variety of methods for channel selection. However, so far, no uniform conclusion has been reached. We summarized 10 articles about channel selection and found that the results for the optimal EEG channel obtained by each article had certain differences. Some of the studies used automatic feature selection methods such as relief, floating generalized sequential backward selection (FGSBS), sequential forward selection (SFS) and evolutionary computation algorithms; some used multiple experiments to verify channel performance; and some researchers manually selected EEG channels based on previous experience. We confirmed the selected EEG channels in this paper by summarizing their commonalities and the physiological basis of the brain. It is worth noting that most of the EEG channels chosen in the literature are located in the lower frontal lobe. There were more than five articles that selected F3, F4, Fp1, Fp2, F7 and F8 [19,33,34,35,36,37]; some articles chose AF3 and AF4 [30,34,35,36]; seven papers selected the T7 and T8 channels located in the temporal lobe; and three papers selected the P7 and P8 channels located in the parietal lobe [10,19,30,33,35,36,38]. In this paper, we manually selected the EEG channel based on previous experience and certain theoretical knowledge. Furthermore, according to the literature [39], it is known that the degree of activity of the left and right frontal lobe can indicate the degree of positivity. This can achieve a high recognition accuracy rate, so this experiment used 8 frontal lobe electrodes (F3, F4, Fp1, Fp2, F7, F8, AF3 and AF4). The alpha wave in the parietal lobe has a strong relationship with people’s relaxation and tension, so two parietal electrodes (P7 and P8) were used in this experiment. Since the sample was a music video, human vision and hearing were both stimulated and affected. Thus, it was necessary to choose the electrodes of the temporal lobe auditory cortex and the occipital lobe visual cortex [40]. Therefore, we chose not only T7 and T8, but also O1 and O2. In addition, the CZ central electrode was selected for our experiments. In summary, 15 channels were selected in total. The specific names of the channel electrodes are shown in Table 1.

### 2.3. Z-Score Normalization

Normally, we need to standardize data to eliminate the influence of dimensions before modeling. If the unstandardized data are directly trained, they may cause the model performance to not be ideal [41]. This is because the model learns too much for variables with large values and not enough for variables with small values. Data standardization methods include min–max normalization and z-score normalization. The z-score method can handle outliers, in contrast to the other methods. As a result, z-score normalization was used in this paper. The specific formulas are:(1)$$\sigma =\sqrt{\frac{1}{N}{\sum_{i=1}^{N}\left(x_{i}-\mu \right)}^{2}}$$
(2)$$z=\frac{x-\mu }{\sigma }$$

### 2.4. Feature Extraction

The main purpose of feature extraction is to extract the information from EEG signals that can significantly reflect the emotional state. The features were extracted from the 15 selected channels. In the process of frequency domain feature extraction, the authors of [42] mentioned that high-frequency channels provide a greater contribution to emotion recognition. We only selected the alpha, beta and gamma bands to extract features, which greatly reduced the amount of calculation. The features are shown in Table 2. A total of 249 dimensional features were extracted.

#### 2.4.1. Time-Domain Analysis

Because the time-domain waveform contains all EEG information without any loss of information [12], the time-domain features were extracted, such as the mean, the standard deviation for the original signal and the signal after the first difference (1^ST^). The formulas involved are shown below, where x is the EEG time-domain signal data, and *T* is the length of the signal. In this study, the value of *T* was 1280.
(3)$$\mu_{x}=\frac{1}{T}\sum_{t=1}^{T}x(t)\sqrt{b^{2}-4ac}$$
(4)$$\sigma_{x}=\sqrt{\frac{\sum_{t=1}^{T}\left(x(t)-\mu_{x}\right)}{T}}$$
(5)$$\delta_{x}=\frac{1}{T-1}\sum_{t=1}^{T-1}\left|x(t+1)-x(t)\right|$$
(6)$$\gamma_{x}=\frac{1}{T-2}\sum_{t=1}^{T-2}\left|x(t+2)-x(t)\right|$$

#### 2.4.2. Frequency-Domain Analysis

The differences between EEG components are mainly reflected in the frequency-domain features, which are indispensable parts of signal analysis. The EEG signal is composed of 5 different frequency bands, including delta (1–4 Hz), theta (4–8 Hz), alpha (8–14 Hz), beta (14–30 Hz) and gamma (30–47 Hz) [43]. However, researchers have shown that low-frequency bands contribute less to emotion recognition and that alpha, beta and gamma bands are more representative [44]. As a result, we extracted the frequency-domain features of these three frequency bands. The Butterworth filter was used to limit the frequency band to alpha, beta and gamma frequency bands [45]. Then, we used Fourier transform (FT) to transform the filtered signal into the frequency domain and extracted the features [43]. The frequency spectra of the three frequency bands after filtering are shown in Figure 2. After Butterworth filtering, this experiment calculated the frequency band energies of the alpha, beta and gamma bands as frequency-domain features [46].

In addition, studies have shown that asymmetric ratios play an important role in emotion classification because these ratios can reflect changes in the left and right hemispheres of the brain [47,48]. In particular, some emotions affect only the left side of the brain, while others affect only the right hemisphere [49]. We extracted some rational features from the power of the alpha and gamma bands. We used the F3, F4, AF3 and AF4 channels for *RASM*, because differences are obvious in these channels. The following formula was used to calculate the *RASM* [50].
(7)$$RASM=\frac{P_{left}}{P_{right}}$$
where $P_{left}$ and $P_{right}$ represent the power of the channels on the left and right hemispheres of the brain. In addition, we are unsure as to whether emotions may cause differences in power across large areas of the brain. We also extracted the total power of the features of channels outside the frontal lobe region, including P7, T7, O1, P8, T8, O2 and CZ. The results of our feature selection may be able to determine whether a particular feature is beneficial to emotion recognition. The detailed feature names are shown in Table 2.

#### 2.4.3. Nonlinear Dynamics

Some studies have shown that the human brain is a complex nonlinear dynamic system. Later on, researchers tried to use nonlinear dynamics to analyze EEG signals [51]. There are two common methods, namely chaos theory and information theory. The following features were extracted in this paper.

##### Entropy Features

The human brain is an extremely complex system, and EEG signals are random and nonlinear. Therefore, we needed to conduct not only linear research on EEG signals, but also nonlinear analysis [30]. With the development of nonlinear dynamics, the entropy feature has become an object of concern, as it describes the nonlinear characteristics of EEG signals [52,53]. Entropy is usually used to show the complexity of objects and has been widely used by researchers in recent years. Many entropy features were extracted from the selected channels, such as the sample entropy [54], approximate entropy, differential entropy [46] and wavelet entropy [55]. The relevant formulas for the entropy features are as follows:(8)$$\begin{array}{l} SampEn(m,r,N)=-\ln(\frac{A^{m}(r)}{B^{m}(r)})\\ ApEn=\Phi^{m}(r)-\Phi^{m+1}(r)\\ DE=\frac{1}{2}\log(2\pi e\sigma_{i}^{2})\\ WE=-\sum_{j}p_{j}\ln(P_{j}) \end{array}$$

##### Lyapunov Exponent

In the process of information extraction, it is very important to analyze and distinguish the working state of a system [56]. Determining the maximum Lyapunov exponent is considered to be the most effective method to identify the state of a system. Therefore, in this experiment, we obtained the value of the Lyapunov exponent for the nonlinear time series of EEG signals. The relevant formula is as follows:(9)$$\lambda =\lim\frac{1}{n}\sum_{k=0}^{n-1}\ln\left|\frac{df(x_{k},\alpha )}{dx}\right|,n\to \infty$$

##### Fractal Dimension

The fractal dimension (FD) can directly evaluate the complexity of time series in the time domain [57]. It has received extensive attention as a successful feature. In EEG signals, various algorithms are used to calculate the FD value. For example, Sevcik’s method, fractal Brownian motion, box counting and the Higuchi algorithm have been employed. As we all know, the result of the latter is closer to the theoretical FD value. Finally, the Higuchi algorithm was used to calculate the FD value, and the formula is as follows:(10)$$FD_{x}=\frac{\left(L(k)\right)}{-\ln k}$$

##### Hurst Exponent

The Hurst exponent is used as an indicator to judge whether the time series data follow a random-walk or a biased random-walk process. There are many methods used to calculate the Hurst exponent. We used the R/S method, and the formulas are as follows:(11)$$\begin{array}{l} \bar{RS}=\frac{1}{g}\sum_{i=1}^{g}RS_{i}\\ s_{RS}=\sqrt{\frac{1}{g-1}{\sum_{i=1}^{g}\left(RS_{i}-\bar{RS}\right)}^{2}} \end{array}$$

The slope of the straight line obtained after calculating a linear regression for the corresponding data is the Hurst exponent, denoted by H. The final calculated value of the Hurst exponent mainly displayed the following characteristics:(1)When 0 < H < 0.5, it indicates that the time series has long-term correlation, but the general trend in the future is opposite to the past, that is, anti-persistence.(2)When H = 0.5, it indicates that the time series is random and uncorrelated, and the present will not affect the future.(3)When H > 0.5, it indicates that the time series has long-term correlation characteristics, i.e., there is continuity in the process.(4)When H = 1, it means that the future can be predicted by the present.

##### Hjorth Parameters

The Hjorth parameters are a method to describe the general characteristics of an EEG trace using several quantitative terms that can be used in EEG studies. In this experiment, for example, only the mobility and complexity features were extracted [58]. The relevant formulas are as follows:(12)$$Mobility:M_{\xi }=\sqrt{\frac{\mathrm{var}(\dot{\xi }(t))}{\mathrm{var}(\xi (t))}}$$
(13)$$Complexity:C_{\xi }=\frac{M(\dot{\xi }(t))}{M(\xi (t))}$$

### 2.5. Feature Selection

In emotion recognition, high-dimensional features may cause a dimensional disaster and increase the amount of calculation. Therefore, the dimensionality reduction of EEG features is an important step in emotion recognition. Selecting an effective feature reduction and selection algorithm can not only increase the speed of model training, but also improve the accuracy and generalization of the model. In this paper, we adopted the method of combining mutual information and SFFS. SFFS is a feature-selection algorithm which requires a large amount of calculation. Therefore, the mutual information feature-selection algorithm was adopted before the SFFS algorithm to filter some features. The remaining features were used for further feature selection through SFFS.

#### Mutual Information

The concept of mutual information originated from information theory and is used to represent the relationship between information. Mutual information can calculate the information shared between the independent variable and the dependent variable and detect the nonlinear relationship between features [59]. Assuming that X and Y are discrete random variables, the mutual information calculation formula is:(14)$$I(X;Y)=\sum_{y\in Y}\sum_{x\in X}p(x,y)\log(\frac{p(x,y)}{p(x)p(y)})$$
where p(x,y) is the joint probability distribution function of X and Y, and p(x) and p(y) are the marginal probability distribution functions of X and Y, respectively.

### 2.6. Classifiers

The accuracy of emotion recognition depends largely on the classifier. In order to ensure accuracy and verify the effectiveness of the dimensionality reduction algorithms, we briefly introduced the following three classifiers, namely k-nearest neighbors (KNN), support vector machine (SVM) and random forest (RF).

#### 2.6.1. KNN

The KNN algorithm is a relatively simple algorithm currently used in data classification. When KNN is used for classification, the predicted result of the sample is the same as the result of its closest sample [60]. The principle of KNN is that when we want to determine the category of some unknown samples, we need to use some known sample categories as a reference and calculate the distance between the unknown and known samples. When predicting a sample’s category, we select k known samples that are closest to this sample, and then the category of this sample is the same as the category with the largest number of k known samples.

#### 2.6.2. SVM

The SVM classifier is a model for binary classification. It distinguishes samples by finding an optimal decision boundary. The decision boundary divides the linear samples so that the interval between the divided samples is maximized [38]. The data samples in this study comprised nonlinear data. In this work, polynomial and radial basis function (RBF) kernels were used with the SVM classifier to evaluate the classification performance. Each sample was mapped to an infinite-dimensional feature space, so that the original linearly indivisible data became linearly separable.

#### 2.6.3. RF

RF is an algorithm that integrates multiple trees through the idea of ensemble learning. Its basic unit is the decision tree. Its essence belongs to a large branch of machine learning—ensemble learning [61]. From an intuitive point of view, each decision tree is a classifier (assuming that it is a classification problem). Then, for an input sample, N trees will have N classification results. Random forest integrates all classification voting results and designates the category with the most votes as the final output. This is the simplest bagging concept. Compared with other classification algorithms, RF has excellent accuracy. It can also process high-dimensional features and effectively process big data.

### 2.7. Multi-Classifier Fusion

The main idea of Multi-Classifier Fusion is to fuse the outputs of multiple classifiers and use them as new features to improve the classification accuracy. Before explaining the Multi-Classifier Fusion method in detail, let us briefly describe the idea of the attention mechanism, because the concept of Multi-Classifier Fusion stems from the attention mechanism, which is widely used in deep learning.

Most studies have proven that the attention mechanism can improve the interpretability of neural networks. The attention model was originally used for machine translation, but later on it gained a large number of applications in the fields of natural language processing, statistical learning, speech and computer science [62]. The attention mechanism can give a neural network the ability to focus on a subset of its features, and it can select specific inputs. Analogous to the human brain, it can filter out unimportant information according to the every-day importance of the information—in fact, this ability is called attention. At present, most attention models are attached to the encoder–decoder framework, as shown in Figure 3 [63]. The left side is the encoder–decoder architecture. The signs h(1) to h(3) and s1, s2 represent the hidden states of the encoder and decoder. The input x is encoded to obtain the h vector and then decoded to obtain the corresponding y. On the right is the attention model based on the encoder–decoder structure. The attention module in the network structure is responsible for automatically learning the weight of attention aij, which can automatically capture the correlation between h and s. Then, these attention weights are used to construct the content vector c. The content vector is the weight sum of all hidden states of the encoder and the corresponding attention weights. Finally, the vector is passed to the decoder.
(15)$$c_{j}=\sum_{i=1}^{T}aij*h(i)$$

In essence, the attention mechanism is designed to perform a weight summation. In fact, the attention mechanism selects the important information from a large amount of information and focuses on it by ignoring most unimportant information. The process of focusing is reflected in the calculation of weight coefficients. The larger the weight, the more focus is placed on its corresponding value. In addition, the weight represents the importance of information.

In the emotion classification experiment in this paper, we referred to the idea of the attention mechanism and applied it to our Multi-Classifier Fusion model in machine learning [64]. The specific process of Multi-Classifier Fusion in cross-subject emotion recognition can be briefly represented by Figure 4. The label X_ori represents the 249 originally extracted dimensional features, and X_select represents the features retained after feature selection. The selected features are input into the KNN, SVM and RF models for classification. This process is analogous to the encoding process in the attention mechanism. After classification, the prediction result of each sample is given its corresponding probability. The output probability can be regarded as a kind of weight. We added the weight outputs from the different classifiers as new weight features. This process is analogous to the method of obtaining c, which is illustrated in Figure 3. In other words, the fusion of the classifiers mainly uses the output probability of the result as the weight for the corresponding processing. Finally, the new weight features and the original features (X_ori) extracted from the selected channels were sent to the classifiers for classification.

## 3. Results

In this experiment, we extracted the time domain, frequency domain, and nonlinearity features from 15 specific EEG channels. We combined different feature selection algorithms for feature selection and retained the features that contributed better to using Multi-Classifier Fusion for classification. In addition, we also paid attention to a problem in the experiment. We extracted different features for different channels. However, the values of these features often have different ranges, a situation which often affects the results of data analysis. Therefore, we used z-score normalization for feature normalization before feature selection.

### 3.1. Evaluation Indices

The data used in this experiment were from the DEAP dataset. We used the hold-out method and leave-one-out cross-validation to conduct the experiments. For the hold-out method, we selected the data of 27 individuals as a training set and the data of 5 individuals as a test set. We cut the EEG signals to expand the samples. The final training set comprised 4320 samples, and the test set comprised 800. For the leave-one-out cross-validation, only one individual’s data were used as a test set in each experiment, and the average value was determined as the final result. In addition, the ultimate goal of this experiment was to minimize the number of features used for classification while ensuring as high a classification accuracy as possible. At the same time, the performance of the optimal feature set and the Multi-Classifier Fusion method proposed in this paper could be verified. Therefore, the classification accuracy was the evaluation index that could intuitively evaluate the rationality and effectiveness of the optimal feature subset and the Multi-Classifier Fusion method. Furthermore, for a more effective comparison, we considered indicators such as accuracy, precision, F1-score and recall. The classification indicators are defined as follows:(16)$$Acc=\frac{TP+TN}{TP+TN+FP+FN}$$
(17)$$Precision=\frac{TP}{TP+FP}$$
(18)$$Recall=\frac{TP}{TP+FN}$$
(19)$$F1-score=2\times \frac{precision\times recall}{precision+recall}$$
where *TP*, *TN*, *FP* and *FN* are true positive, true negative, false positive and false negative, respectively.

### 3.2. Determination of Feature-Selection Method

In this paper, we extracted high-dimensional features in the time domain, frequency domain and nonlinear domain. However, high-dimensional features may cause dimensional disasters. In this section, the feature-selection methods that were attempted include SBS, SFS, SFFS, principal component analysis (PCA), correlation and mutual information. We found the optimal feature-selection method by exploring the classification accuracy based on a single feature-selection method and the combination of two feature-selection methods.

First, we determined the classification accuracy obtained by retaining different numbers of features. According to the results, the influence of the number of features on the classification performance could be further verified. We used the SBS method that has been previously implemented in the literature [28], which includes SVM as a classifier for screening features. In addition, we used KNN for comparison. Figure 5 shows the classification accuracy of different numbers of features.

In Figure 5a,b, the accuracy rates of both methods are affected by the excessively high feature dimensions, and SVM is better than KNN in terms of accuracy. However, it can be clearly seen from Figure 5c that it took 52,108.322 s to use KNN for feature selection, while it took 249,975.318 s to use SVM. In addition, there were two parameters that needed to be adjusted in SVM, while KNN contained only one parameter. This also means that, compared with KNN, the adjustment of parameters is a major problem when using SVM, which requires a lot of time. Therefore, we planned to implement KNN as the classification algorithm in the selector.

When SBS selects features it starts from the full set of features. It removes an unnecessary feature from the feature set each time, so that the evaluation function value reaches the optimal value. Therefore, its disadvantage is that it can remove features but not add features. Similarly, SFS can add features but not remove features. Both of these methods can easily fall into local optimal values. Later on, SFFS was proposed in view of the shortcomings of the two abovementioned algorithms. This method starts from an empty set, adds a subset, and then removes the subset from the selected features to optimize the evaluation function. This algorithm makes up for the shortcomings of the first two algorithms.

Then, in order to verify the effectiveness of these three feature-selection algorithms, we implemented them for feature selection. We applied SVM to train and classify the test set according to the features selected by the different methods. Taking the retained feature dimension of 100 as an example, the results were as shown in Table 3.

Table 3 shows the classification accuracy of positive emotions and negative emotions in three different feature sets. For these three feature-selection methods, SFFS had the highest classification accuracy, indicating that the SFFS method is more effective than the other methods. In addition, the precision and f1-score values of the SFFS method were higher than the other methods. Therefore, we conducted further research based on the SFFS method. However, SFFS adopts the technique of traversing every feature subset in its feature selection, which leads to a large computational cost. In order to alleviate this problem, we planned to adopt other feature selection methods to filter out most redundant features before using SFFS.

In this section, we study the classification accuracy based on the SFFS method combined with different feature-selection methods, including principal component analysis, correlation feature selection and the mutual information method. Table 4 displays the comparison of classification performance between the combined feature selection methods and individual methods. Because the parameters of the different methods varied, the retained features could not be consistent after the feature selection. We set up a similar number of features for comparison.

Table 4 shows that, compared with a single feature-selection method, the combined feature-selection method improved the classification accuracy, precision, recall and f1-score to a certain extent. On the other hand, the combination of different methods will achieve different classification effects.

PCA is a feature dimensionality reduction algorithm. It can be used to extract the main feature components of a dataset [60]. As shown in Table 4, the original features were reduced to 130 dimensions and sent to the classifier, and the accuracy rate of the classifier was 0.6775. When the 130 dimensional features were sent to SFFS for further feature selection, reducing them to 80 features, the classification accuracy rate reached 0.67875. This again verifies that effective feature selection can improve the classification accuracy.

Correlation feature selection is mainly intended to filter features by calculating the correlation between the features and the targets. If the correlation between the features and the targets is 1, it means that there is a high degree of positive correlation between them. If it is −1, there is a highly negative correlation. In this experiment, the parameter was set to 0.93, which meant that the features with a correlation above 0.93 were retained. One hundred and thirty-one dimensional features were finally retained. The classification results are shown in Table 4.

The parameter of mutual information feature selection was set to 0.5, so as to retain half of the original features. From the classification results, it can be seen that the 124 retained dimensional features could be sent to the classifier to achieve a higher accuracy than PCA and correlation feature selection. Combining MI and SFFS achieved the largest recall and f1-score values compared to the other feature-selection methods. This verifies the effectiveness of mutual information combined with SFFS feature selection.

### 3.3. Parameter Setting of Feature Selection

Table 4, above, verifies the effectiveness of mutual information combined with SFFS. However, the parameter of mutual information and the number of features retained after SFFS affect the performance of the classification. Different parameters will cause different combinations of features to be retained. In addition, different numbers of features will also lead to different classification effectiveness for emotion recognition. Therefore, in this section, we try to set different parameters and retain a different number of features to evaluate the accuracy of recognition.

As shown in Table 5, the highest accuracy rate achieved was 0.6825, when the parameter was set to 0.5 and the number of retained features was 70. These results show that when the mutual information parameter was 0.5, the model was able to achieve the highest classification accuracy. The values for the precision and f1-score reached 0.6230 and 0.7590, respectively, which were higher than the results obtained by other parameters. When the mutual information parameter was set to less than 0.5, such as when it was set to 0.38, 0.4 and 0.45, the highest accuracy rate was 0.68. This relatively poor accuracy may be caused by the loss of features favorable for emotion recognition. In the end, we determined that the parameter of mutual information should be set to 0.5. Our ultimate goal was to use as few features as possible to achieve the highest accuracy. In order to obtain the optimal feature set, KNN, SVM and RF were used individually for classification, and the classification results are shown in Figure 6.

As can be seen from Figure 6, when the number of selected features was 65, the accuracy rate was better. The classification accuracy rate of RF reached 0.695. In addition, there were common patterns across each group of features. RF showed the best classification performance, followed by SVM, while KNN had no obvious advantage. Therefore, SVM and RF were used for classification verification in the subsequent experiments.

### 3.4. Application of Multi-Classifier Fusion

#### 3.4.1. Hold-Out Method

In order to further improve the accuracy of classification, we adopted the method of Multi-Classifier Fusion introduced above. This method stems from the idea of using weights in the attention mechanism. Sixty-five features were kept and sent to SVM, KNN and RF for classification. We output the probabilities of the classification results of SVM and RF and added them together as a new set of weight features. The new features together with the original features were sent to classifiers for further classification, and the results are shown in Figure 7 and Table 6.

As shown in Figure 7, the classification accuracy of KNN (k = 41) reached 0.68375 after applying the new model. The classification accuracy of SVM (C = 0.7, gamma = 0.015) reached 0.69625 after applying the new model. The classification accuracy of RF reached 0.70375. The accuracy of these three classifiers was significantly improved. From Table 6, we can see that there was no significant change in the results of KNN, but the indicators of RF were improved. This conclusion verifies the effectiveness of this study to a certain extent.

#### 3.4.2. Leave-One-Out Cross-Validation

However, because the hold-out method was adopted to verify performance in the above experiments, the grouping of the original data had a certain influence on the final classification accuracy, so the results obtained by this method are not authoritative.

In order to make the results more representative, we used leave-one-out cross-validation to further verify the performance. During the experiment, we first used the single-subject samples as the test set for feature selection to retain 65 optimal features. Then, each subject was considered as a test set once, while the remaining subjects were considered for training. The average of all iterations was used as the final result, and the model performance was evaluated for accuracy. In addition, we originally included a total of 32 subjects. During the experiment, we found that there were some problems with the data of 4 subjects, so we only conducted experiments on the remaining 28 subjects.

In this experiment, four different Multi-Classifier Fusion methods were included, namely KNN and RF, KNN and SVM, SVM and RF, and KNN and RF, SVM. The results are shown in Figure 8. Figure 8a shows the accuracy comparison between the accuracy of 65 features and the result after adding the output probabilities of KNN and RF as a new feature for classification. Figure 8b–d show similar comparisons, but for different classifiers.

The dotted line represents the accuracy of the original method. From the four figures, we can conclude that combining the probabilities of different classifiers can improve the classification performance to a certain extent. However, the effect of combining KNN and SVM in this experiment was slightly worse than the other methods.

In order to reflect the classification performance more clearly, we obtained the average of all the subjects’ classification results for comparison. The results are shown in Figure 9 and Table 7. From Figure 9, we can clearly see that the accuracies of all four kinds of model were higher than the original model. After using the proposed Multi-Classifier Fusion model, the classification accuracy of SVM was improved by 3%. Notably, when the probability of KNN and RF was combined, the classification effect of RF reached 0.7361. In Table 7, we averaged the results for each classification indicator for all subjects. We compared the classification performance between the original model and the four kinds of Multi-Classifier Fusion model. In the table, the selected methods are listed in the column on the left side. For example, the label ‘Original’ represents the original method, and KNN + RF, KNN + SVM, RF + SVM and KNN + RF + SVM are the four Multi-Classifier Fusion methods. The top indicators were the precision, recall and f1-score output by the SVM, KNN and RF classifiers. We found that the highest values of the precision, recall and f1-score indicators appeared with very high frequency in the results of the four Multi-Classifier Fusion models.

## 4. Discussion

The Multi-Classifier Fusion model is proposed based on the output probabilities of the classifiers in this paper. This method improves the accuracy of classification to a certain extent. The dataset used in this paper was DEAP. EEG signals were recorded by 32 brain electrodes while each subject watched a 60 s video. We only interpreted the last 40 s of the data for processing. This prevented the first 20 s of data from interfering with the accuracy of the emotion recognition, because the subjects had not yet entered the corresponding emotional states in the first 20 s. In this paper, only 15 EEG channels are selected, which greatly reduced the amount of calculation required. In addition, for this experiment we cut the data every 10 s, which greatly expanded the amount of data. Therefore, the dimensions of the final data used for feature extraction were 32 (subjects) * 15 (channels) * 5120 (EEG data samples). The flow chart of this experiment is shown in Figure 10.

This experiment included the stages of data preprocessing, feature extraction, feature selection and classification. The feature extraction of EEG signals is an important part of analyzing EEG characteristics. Most of the previous literature has verified the effectiveness of a certain feature, but no research has clearly shown which feature combinations are optimal for emotion recognition. Therefore, the way to improve the classification accuracy in this paper was to extract as many favorable features as possible and then perform feature selection. Ultimately, we expected to obtain the optimal feature set. When extracting the frequency domain features, we only focused on the three high-frequency bands of alpha, beta and gamma. In addition, we also obtained the RASM features of the EEG signals to represent the difference between the left and right hemispheres of the brain. Besides the time domain and frequency domain features, entropy features and other features such as FD were also extracted to represent the nonlinear characteristics of EEG signals. The entropy feature was mainly derived from information theory. In information theory, entropy is thought to represent the amount of information. In EEG analysis, entropy can be used to describe the complexity and regularity of EEG signals. In the end, a total of 249 dimensional features were collected.

After feature extraction, we used the SBS feature-selection method mentioned in the literature [28]. From the results shown in Figure 5, we found that the classification accuracy was not directly proportional to the feature dimensions because of the existence of redundant features. This phenomenon has been mentioned in the literature [65]. In the process of feature selection, the idea of SBS is to continuously remove features to make the evaluation function optimal. SFS is similar to SBS. However, both methods have certain limitations. The SFFS method was proposed by a later study; this method can make up the shortcomings of SBS and SFS. SFFS can eliminate and add features to make the evaluation function optimal. This paper used all three methods to conduct simple experiments, and the results are shown in Table 3. We can see that the feature classification performance of SFFS was better, which is consistent with the idea proposed by [66]. However, although the selected feature classification performance of SFFS was better, it took a long time. To solve this problem, we chose a feature-selection method to reduce the feature dimensions before employing SFFS. In this paper, the PCA, correlation and mutual information feature-selection methods were chosen for comparative experiments. From Table 4, we can see that the classification accuracy of the combined feature-selection methods was better than a single feature-selection method. For example, the classification accuracy rate after using PCA for dimensionality reduction alone was 0.6775, while the classification accuracy rate after combining PCA and SFFS was slightly improved to 0.67875. What is more, we found that the method of combining mutual information with SFFS achieved the best classification effect. Mutual information performed better than PCA and correlation. Table 4 demonstrates the effectiveness of mutual information feature selection compared with the other feature selection methods. The concept of mutual information comes from information theory. We applied information theory to emotion recognition and demonstrated that it has certain advantages in computational biology. Therefore, the method of combining MI and SFFS was chosen in this paper.

When using the MI method, we needed to manually adjust the parameters to initially filter the features. As shown in Table 5, we chose four different parameter values for selection. The final results show that when the parameter was set to 0.5, the highest accuracy rate was 0.6825. We can infer that the smaller the parameter, the lower the number of features that are retained. This may also result in useful features being filtered out, thereby reducing the accuracy. After determining the parameters, we also needed to consider the number of features retained after using the SFFS feature-selection method. We used SVM and RF to conduct experiments at the same time, and the results are shown in Figure 6. The best accuracy appeared when the number of retained features was 65. Therefore, after a series of experiments, the effectiveness of the MI method combined with the SFFS method was verified.

For a more intuitive analysis of the 65 selected features, we output and observed the selected features. The results of the analysis according to the number of features extracted from each channel are shown in Figure 11a. We found that each channel contained at least two retained features, which proves the validity of the originally manually selected channels. There were a total of 10 features from the CH4 channel, and the CH11 and CH28 channels retained 6 and 7 features, respectively. There were obviously more features extracted from these three channels than the other channels. This phenomenon shows that these three channels play a necessary role in emotion recognition experiments. However, this paper is focused on selecting the optimal feature set and verifying the effectiveness of the Multi-Classifier Fusion method in the experimental process. Therefore, this study did not use the channel selection algorithm to select the EEG channel. Instead, features were manually selected based on previous experience. This was also a shortcoming of this paper. In the future, researchers could combine a certain channel selection algorithm for further improvement.

The results of the analysis carried out according to the number of extracted features are shown in Figure 11b. A total of 25 statistical features were retained, which accounted for a large proportion of all the retained features. Among them, the most preserved feature of the EEG was the mean feature—there were 15 in total. The results are consistent with the conclusions drawn in the literature [30], i.e., that statistical features are the most meaningful features that characterize brain emotions. Secondly, the complexity and FD contributed to the accuracy of emotion recognition, since they can both better describe the nonlinear characteristics of EEG signals.

After the above series of experiments, the highest classification accuracy rate we achieved was 0.695. In order to further improve the accuracy, a new idea was proposed, which was to use the output probabilities of the classifiers as new features for classification. This proposition was based on the weight idea in the attention mechanism. We can also understand it in the following way: in a situation where one classifier is accurate and the other classifier is incorrect when performing classification, adding the corresponding probability weights of the two classifiers can reduce the probability of error to a certain extent. As shown in Figure 7 and Figure 9, the new method did show good performance, with the classification accuracy reaching 0.73.

In addition, cross-subject research using the DEAP dataset has gradually increased in recent years. Each new finding lays the groundwork for future research. Similar works are as follows. In [58], the experimental results show that by using automatic feature-selection methods in DEAP, the average accuracy reached 0.59. In [28], adopting the ST-SBSSVM method and using 32 EEG channels improved the classification accuracy to 72%. As shown in Figure 12, only 15 EEG channels were used in this paper, and the proposed method showed a good performance. Its accuracy is comparable to the cross-subject emotion recognition accuracy using the DEAP dataset in similar studies. In summary, the method proposed in this research is effective for cross-subject emotion recognition.

With the development of deep learning, we find that more and more people are turning their attention to deep learning applications. Although deep learning can indeed achieve good results, it also requires a lot of data. In some research fields, there are not sufficient data, which limits the application of deep learning. At this point, we should try to focus on improving the accuracy of the traditional machine learning algorithm. To a certain extent, traditional machine learning has the advantages of easy implementation and a fast calculation speed. In this paper, we absorbed the idea of the attention mechanism and proposed a Multi-Classifier Fusion model. Experiments were conducted based on this model, and the results also verified the effectiveness of this model. In addition, the model proposed in this paper is actually a new way of thinking. Future researchers could consider trying this approach in their fields. Perhaps this approach could further improve the classification accuracy of research in other fields.

## 5. Conclusions

In this study, we introduced the concept of weight coefficients from the attention mechanism and proposed a Multi-Classifier Fusion model. This model contributed to the improvement of cross-subject emotion recognition accuracy. We chose the DEAP dataset to test the performance of the model. In the process of emotional recognition, data cutting was employed to expand the samples, because sufficient samples were needed to train the model more effectively. In addition, not all channels are beneficial to emotion recognition. Selecting a subset of the channels can reduce the computational cost. When extracting features to express the characteristics, the time domain, frequency domain and nonlinear features were extracted. By comparing the results of combining different feature-selection methods, we found that the performance of combining MI and SFFS was better than the other methods. In the test process, SVM, KNN and RF were used as classifiers to calculate the accuracy of emotional classification. The average emotion recognition accuracies reached 0.6789, 0.688 and 0.7133. Finally, the classification performance was optimized by applying a Multi-Classifier Fusion method. The best classification accuracy was obtained by adding the output probabilities of different classifiers as weight features. The average emotion recognition accuracies of the proposed scheme reached 0.7089, 0.7106 and 0.7361. This shows that the proposed method for cross-subject emotion recognition improves performance when compared with existing methods. However, this study used the idea of introducing weights only once in the experimental process. Better results may be obtained if multiple iterations are performed or if multiple classifiers’ output probabilities are stacked.

## Figures and Tables

**Figure 1 entropy-24-00705-f001:**
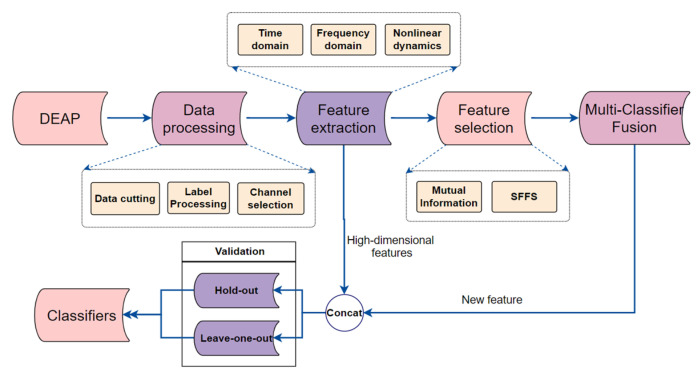
Picture of the overall proposed method. The research sequence of this paper is shown in the figure. DEAP is a multi-modal open dataset containing 32 EEG channels and multiple other physiological signals. We acquired the DEAP dataset; performed data preprocessing, feature extraction and feature selection; and used the Multi-Classifier Fusion method. Finally, we used classifiers for classification.

**Figure 2 entropy-24-00705-f002:**
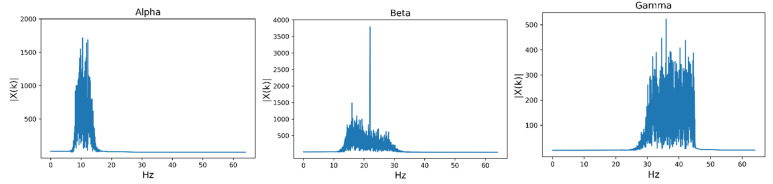
The frequency spectra of the three frequency bands after filtering. The frequency spectra of the alpha, beta and gamma frequency bands after Butterworth filtering are shown in Figure 2. In this study, only three frequency bands were used to extract frequency-domain features.

**Figure 3 entropy-24-00705-f003:**
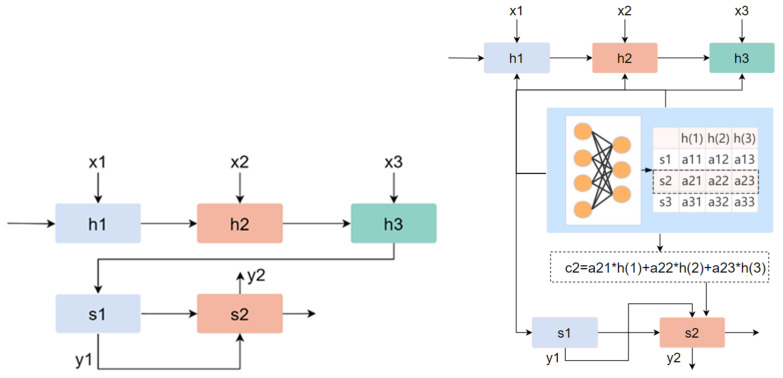
Encoder–decoder architecture and attention model. The left side is the encoder–decoder architecture. Assuming x is the input, after encoding and decoding, we can obtain the final output value y. The right side is the attention model based on the encoder–decoder structure. It shows the process of constructing the vector c.

**Figure 4 entropy-24-00705-f004:**
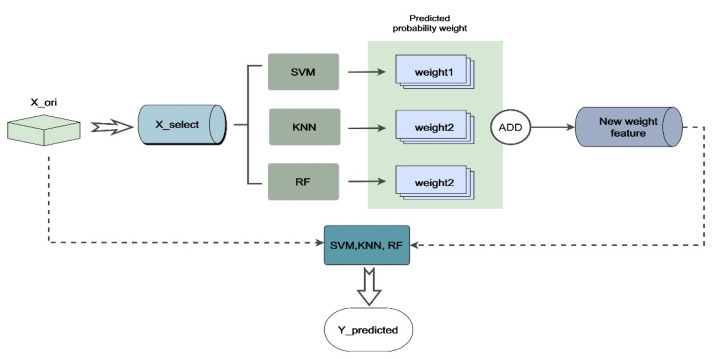
Emotion recognition system flow chart. The label X_ori represents the high-dimensional features during feature extraction, and X_select represents the features selected by the feature selection model. New weight features are obtained by fusing the outputs of KNN, SVM and RF, and then the new weight features are used together with the original features for classification.

**Figure 5 entropy-24-00705-f005:**
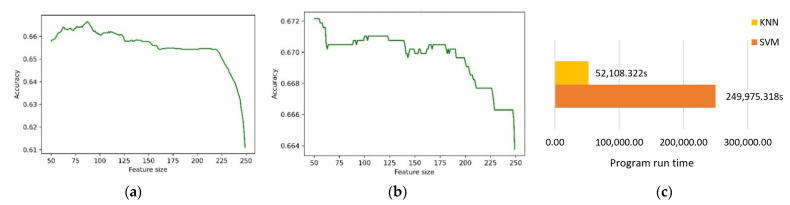
The relationship between feature dimensions and accuracy. When using KNN and SVM for feature selection, the accuracy was affected by the feature dimension. The accuracy of SVM was better than that of KNN, but it took a long time. (**a**) Using KNN as classifier for screening features; (**b**) using SVM as classifier for screening features; (**c**) comparison of run time of KNN and SVM.

**Figure 6 entropy-24-00705-f006:**
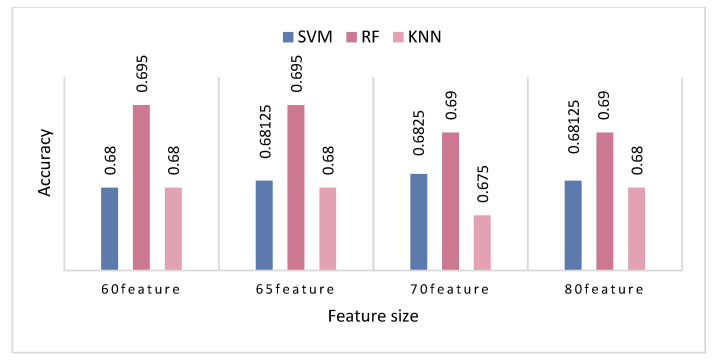
Comparison of classification accuracy of different numbers of features. After feature selection, we retained 60, 65, 70 and 80 EEG features and then sent them for SVM, RF and KNN classification. The results show that the accuracy of the classifiers was optimal when 65 features were retained.

**Figure 7 entropy-24-00705-f007:**
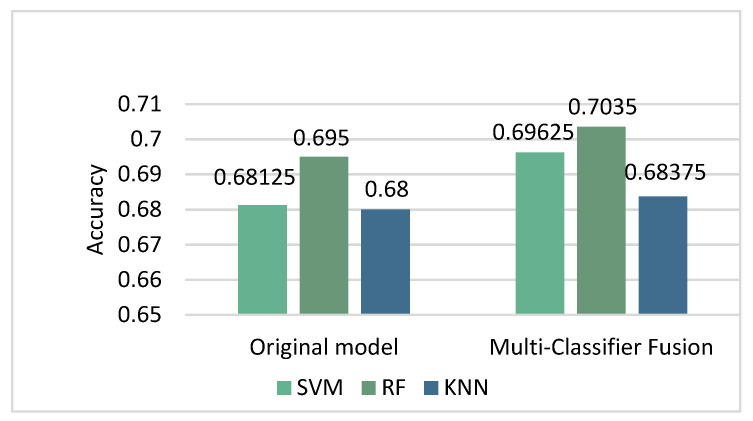
Comparison of the accuracy of original model and Multi-Classifier Fusion model. The figure shows the result comparison between the original model and the Multi-Classifier Fusion model. We selected SVM, KNN and RF for verification. The results show that the accuracy of the classifiers was improved after using Multi-Classifier Fusion.

**Figure 8 entropy-24-00705-f008:**
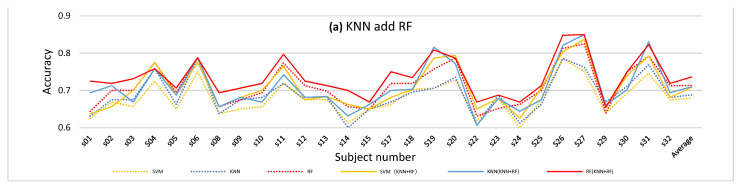
The comparison of the original method and four kinds of Multi-Classifier Fusion method, which fused different classifiers to obtain new features to improve the accuracy. For example, (**a**) shows the comparison of the classification results after fusion of KNN and RF with the original classification results. The dotted yellow line represents the results obtained initially with SVM classification. The solid yellow line represents the result of SVM classification after fusion of KNN and RF. The dotted blue line represents the results obtained initially with KNN classification. The solid blue line represents the result of KNN classification after fusion of KNN and RF. (**b**) shows the comparison of the classification results after fusion of KNN and SVM with the original classification results. The dotted yellow line represents the results obtained initially with SVM classification. The solid yellow line represents the result of SVM classification after fusion of KNN and SVM. (**c**) shows the comparison of the classification results after fusion of SVM and RF with the original classification results. The dotted yellow line represents the results obtained initially with SVM classification. The solid yellow line represents the result of SVM classification after fusion of SVM and RF. (**d**) shows the comparison of the classification results after fusion of KNN, SVM and RF with the original classification results. The dotted yellow line represents the results obtained initially with SVM classification. The solid yellow line represents the result of SVM classification after fusion of KNN, SVM and RF.

**Figure 9 entropy-24-00705-f009:**
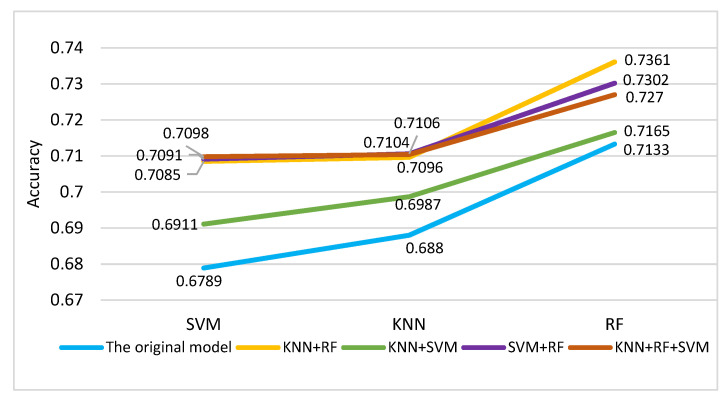
Comparison of the average values of the four Multi-Classifier Fusion methods. The blue lines represent the outputs of the three classifiers for the original model. The remaining four lines are the results of the four Multi-Classifier Fusion methods. We found that the output results of the Multi-Classifier Fusion method were all better than the results of the original method.

**Figure 10 entropy-24-00705-f010:**
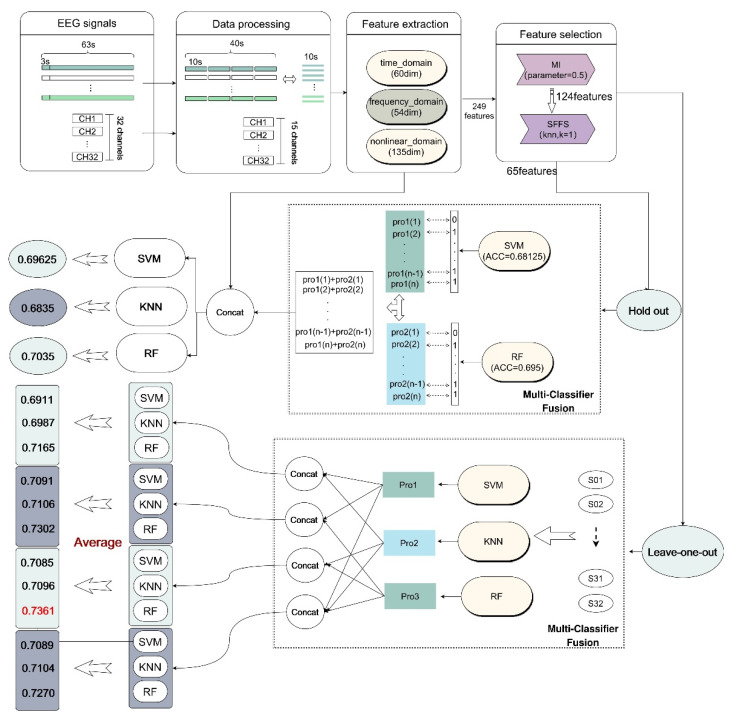
The process of emotion recognition across subjects in the DEAP dataset. The figure describes the specific methods used in the research process of this article. For data preprocessing, we cut the data and selected 15 EEG channels. Then, we extracted high-dimensional features. In feature selection, we used mutual information and SFFS methods to retain 65 EEG features. After using Multi-Classifier Fusion, the leave-one-out method and the hold-out method were used for classification verification. The highest accuracy rate reached 0.7361.

**Figure 11 entropy-24-00705-f011:**
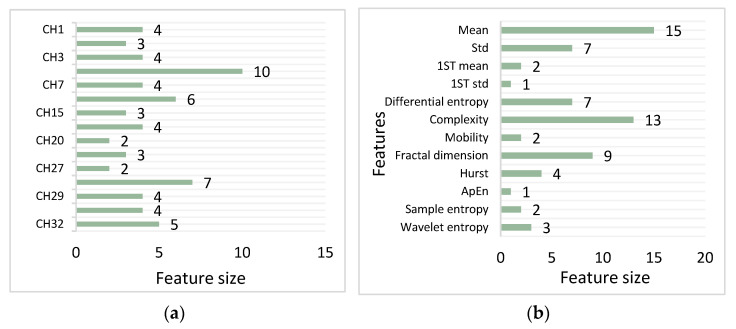
Classification of 65 retained features. (**a**) the distribution of the selected 65 features across the 15 EEG channels. (**b**) the distribution of the selected 65 features across different kinds of features.

**Figure 12 entropy-24-00705-f012:**
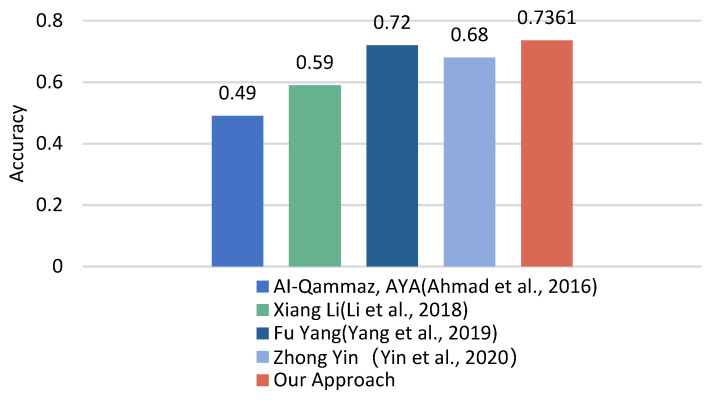
Comparison between this work and previous studies in DEAP. The figure shows the accuracy results of our approach compared with four other research methods. Our classification accuracy is higher than their results and reaches 0.7361.

**Table 1 entropy-24-00705-t001:** 15 selected EEG channels.

Brain Area	Selected Electrodes
Frontal lobe	AF3, AF4, F3, F4, F7, F8, FP1, FP2
Parietal lobe	P7, P8
Temporal lobe	T7, T8
Occipital lobe	O1, O2
Central lobe	CZ

**Table 2 entropy-24-00705-t002:** List of features.

Feature Type	Extracted Features
Time domain	1. Mean 2. Standard deviation 3. First difference mean (1ST mean) 4. First difference standard deviation (1ST std)
Frequency domain	1. Band energy of alpha, beta and gamma 2. Beta/alpha of CH4, beta/alpha of CH27, beta/alpha of CH2, beta/alpha of CH4 3. CH4/CH27′s alpha, CH4/CH27′s beta, CH2/CH29′s alpha, CH2/CH29′s beta 4. Sum of channel powers (CH7, CH11, CH15, CH17, CH20, CH24, CH32)
Nonlinear dynamic	1. Sample entropy 2. Approximate entropy 3. Differential entropy 4. Wavelet entropy 5. Lyapunov exponent 6. Fractal dimension 7. Hurst exponent 8. Hjorth parameter (mobility, complexity)

**Table 3 entropy-24-00705-t003:** Comparison of three feature-selection methods.

Feature-Selection Method	Accuracy	Precision	Recall	F1-Score
SBS	0.6625	0.6093	0.9805	0.7516
SFS	0.6785	0.6205	0.9684	0.7563
SFFS	0.68	0.6211	0.9708	0.7575

**Table 4 entropy-24-00705-t004:** Comparison of the results of combining different feature-selection methods.

Feature-Selection Method	Feature Dimension	Accuracy	Precision	Recall	F1-Score
PCA	130	0.6775	0.6199	0.9660	0.7552
PCA + SFFS	80	0.67875	0.6205	0.9684	0.7563
Correlation (0.93)	131	0.6725	0.6160	0.9660	0.7523
Correlation (0.93) + SFFS	80	0.6775	0.6296	0.9077	0.7435
MI (0.5)	124	0.68	0.6211	0.9708	0.7575
MI (0.5) + SFFS	80	0.68125	0.6217	0.9733	0.7587

**Table 5 entropy-24-00705-t005:** Comparison of classification results for different parameters.

Parameter of MI	Number of Features after MI	Number of Features after SFFS	Accuracy	Precision	Recall	F1-Score
0.35	87	80	0.68	0.6211	0.9708	0.7575
0.4	100	80	0.68	0.6211	0.9708	0.7575
0.45	112	80	0.68	0.6211	0.9708	0.7575
0.5	124	80	0.68125	0.6217	0.9733	0.7587
0.5	124	70	0.6825	0.6230	0.9708	0.7590
0.5	124	65	0.68125	0.6217	0.9733	0.7587

**Table 6 entropy-24-00705-t006:** Classification performance comparison between original model and Multi-Classifier Fusion (hold-out) models.

	Precision		Recall		F1-Score	
	Original	Multi-Classifier Fusion	Original	Multi-Classifier Fusion	Original	Multi-Classifier Fusion
KNN	0.6172	0.6172	0.9708	0.9708	0.7547	0.7547
SVM	0.6217	0.6449	0.9733	0.9126	0.7587	0.7557
RF	0.6377	0.6409	0.9660	0.9660	0.7653	0.7705

**Table 7 entropy-24-00705-t007:** Classification performance comparison between original model and Multi-Classifier Fusion (leave-one-out) models.

	SVM			KNN			RF		
	Precision	Recall	F1-Score	Precision	Recall	F1-Score	Precision	Recall	F1-Score
Original	0.6247	0.8867	0.7408	0.6121	0.9217	0.7459	0.6651	0.8887	0.7563
KNN + RF	0.6654	0.8707	0.7498	0.6199	0.9458	0.7475	0.705	0.8399	0.7616
KNN + SVM	0.6399	0.9045	0.7449	0.6125	0.9529	0.7447	0.6767	0.8462	0.7494
RF + SVM	0.6701	0.8478	0.7439	0.6205	0.9386	0.7456	0.694	0.8442	0.7584
KNN + RF + SVM	0.6677	0.8609	0.7469	0.6205	0.9387	0.7456	0.6929	0.8398	0.7547

## Data Availability

Data are available in a publicly accessible repository that does not issue DOIs. These data can be found at the following address: http://www.eecs.qmul.ac.uk/mmv/datasets/deap/index.html (accessed on 16 November 2020). The data presented in this study are available on request in the Appendix A.

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
