# Peer review of "Multi-Classifier Fusion Based on MI–SFFS for Cross-Subject Emotion Recognition"

_entropy, 2022, doi:10.3390/e24050705_

Round 1
Reviewer 1 Report
Dear Authors,
Please find the attached file for my concerns. Please update the paper and resubmit it.
Best Regards

Author Response
We would like to thank you for your efforts in reviewing our manuscript titled "Multi-Classifier Fusion based on MI_SFFS for cross-subject emotion recognition " (ID: 1662769), and providing many helpful comments and suggestions. Regarding your suggestion, our reply content is in the document

Reviewer 2 Report
This study proposes an emotion recognition system based on electroencephalography signals. The manuscript is easy readable, but it can benefit after adding more technical details, and proofreading. I have some comments as follows:
1- The state-of-the-art about emotion recognition can be discussed better in section Introduction. For instance, several studies about facial expression recognition by using thermal images should be included.
2- The authors should review the sentence on the lines69-71 pp. 2 "The author find that..."
3- In section 2.2 the authors should comment related work that applied automatic EEG channel selection for emotion classification. Furthermore, the authors should explain how the EEG channel selected manually in Table 1m agree with these previous studies.
4- Table 2 is not relevant for readers. I suggest to remove it. A similar action can be carried out with Figure 5, as it shows the same information contained in Figure 6.
5- Figure 2 should be edited by adding the scale values in y-axis for each power spectrum.
6- Feature selection algorithms search best features on a feature set. I cannot understand why the authors selected manually EEG channels for emotion recognition, as well as to extract some features, such as RASM and total power. The authors should discuss about it.
7- Figure 4 can be replaced by other diagram more link to the current research.
8- The authors should provide more information about KNN and SVM, specifying the used k value and kernel, respectively.
9- Other popular public EEG databasets for emotion recognition should be added for evaluation, and compare better with the state-of-the-art.
10- The authors should review carefully the references.
Author Response
We would like to thank you for your efforts in reviewing our manuscript titled "Multi-Classifier Fusion based on MI_SFFS for cross-subject emotion recognition " (ID: 1662769), and providing many helpful comments and suggestions. Regarding your suggestion, our reply content is in the document.

Round 2
Reviewer 2 Report
The presentation of the revised manuscript has been improved, and provide more technical details and information for readers. This version is well organized and easy readable. I have still some minor suggestions:
1- In table 2, the authors should replace the terms: "Time domains" by "Time domain", "Frequency domains" by "Frequency domain", and "Non-linear dynamics by Non-linear dynamic".
2- Please check in Table 7, specially on its columns, if SVM, KNN, and RF should be indicated. It is confusing.
Author Response
Thanks again for your suggestions, we have revised the manuscript according to your suggestions.
